

# Trends in soil organic matter and topsoil thickness under regenerative practices at the University of Washington student farm

Julia E. Macray[1] and David R. Montgomery[1]

Department of Earth & Space Sciences, University of Washington, Seattle, WA, USA

## ABSTRACT

Conventional methods of agriculture, especially tillage, are often accompanied by soil degradation in the form of erosion and organic matter depletion. Regenerative agricultural methods seek to repair soil ecosystems by building topsoil and soil organic matter (SOM), decreasing reliance on chemical fertilizers and increasing both water retention capacity and the diversity and quantity of soil microbial and fungal communities. The University of Washington (UW) student farm is an organic and regeneratively managed site on the UW Seattle campus. Over the past 20 years the farm gradually expanded so locations on the farm encompass both unimproved topsoil and soils managed regeneratively for periods of 5 to 20 years. This arrangement allows a time-trend analysis of soil development under regenerative methods. Measurements of topsoil thickness (defined as the distance from the ground surface to the base of the soil A horizon) and organic matter content were collected across 14 distinct plots on the farm to quantify trends over time and estimate net change in SOM (and soil organic carbon, or SOC). While SOM content weakly increased by 0.5% per year, topsoil thickness exhibited a significant linear increase of 0.86 cm per year. Over a twenty-year period under the management practices of the UW Farm total organic carbon storage in soils, determined using topsoil thickness, density, and SOC content, increased by between 4 and 14 t ha$^{-1}$ yr$^{-1}$. The general increases in topsoil thickness, SOM content, and total soil carbon demonstrate the potential of soil-health-focused practices to help maintain a productive and efficient urban growing space.

# INTRODUCTION

Soils are an integral component of terrestrial ecosystems, holding nutrients, water, and carbon, and fostering communities of fungi and microorganisms that are essential to the function of larger flora and fauna and the ecosystem as a whole. Conventional forms of agriculture that utilize tillage have degraded soil ecosystems across large swaths of the United States, although topsoil erosion and changes in soil organic matter are often dependent on soil type, local climate and hydrology, and other factors that vary from farm to farm (*Baumhardt, Stewart & Sainju, 2015*). In general, however, rates of soil loss on

Corresponding author
David R. Montgomery,
bigdirt@uw.edu

conventionally tilled farms far outpace natural rates of erosion (*Montgomery, 2007*; *Thaler et al., 2022*). Tillage can strip the soil of the A-horizon (*e.g.*, *Thaler, Larsen & Yu, 2021*) and degrade soil organic carbon (SOC) (*Tiessen, Cuevas & Chacon, 1994*) which in turn decreases production capacity, soil moisture retention capacity, and bioavailable nutrients. Since the implementation of the Homestead Act in 1862, approximately 35% of soils across the Corn Belt of the American Midwest have been eroded through the A-horizon, removing roughly 1.4 Pg of carbon from degraded regions (*Thaler, Larsen & Yu, 2021*).

Although an explicit definition of regenerative agriculture has not been widely accepted in scientific literature, it can be broadly considered to encompass varied agricultural methods that produce beneficial changes in soil (*Newton et al., 2020*), particularly by increasing SOC and enhancing soil health through farming systems following the principles of Conservation Agriculture in combining the use of no-till, cover crops, and diverse crop rotations (*Montgomery, 2017*; *Kassam, Friedrich & Derpsch, 2019*). A recent comparison of topsoil health and organic matter content on paired farms found this combination of practices to increase both relative to levels under conventional practices (*Montgomery et al., 2022*), and use of cover crops has been estimated to increase soil carbon by 0.32 t ha$^{-1}$ yr$^{-1}$ (*Poeplau & Don, 2015*). Regenerative agricultural methods that build soil health (*Schreefel et al., 2020*) have also been linked to increased nutrient density in crops (*Montgomery & Biklé, 2022*). While adopting no-till farming can greatly reduce soil erosion (*Montgomery, 2007*; *Kwang, Thaler & Larsen, 2023*), to date only limited data are available on how much and how fast regenerative farming systems using all three techniques—no-till, cover crops, and diverse rotations—can increase soil organic matter, and thereby enhance soil fertility. Here we report an analysis of time-trends in soil thickness and organic matter content based on plots at the University of Washington Student Farm (UW Farm), located at the Center for Urban Horticulture (CUH) on the UW Seattle Campus.

## STUDY AREA

The UW Farm presents a novel experimental space, as a gradual expansion of the area under regenerative cultivation created a natural site for a space-for-time analysis. The CUH site is located in the Union Bay Natural Area, the historical site of the Montlake Dump which was used until 1966. Following closure of the landfill, remediation and restoration efforts removed invasive Himalayan blackberries and placed a cap of gravel and earth on top of the site, encouraging growth of native plant species. A small portion of the UW Farm, now known as "Plot E", was used for one season in 1995 as a space for experimental wheat growth trials, but ground was not broken in the other areas until 2002, when Seattle Tilth Alliance began farming in the area now known as "Plot H". The farm slowly expanded over the next two decades as a student-operated farm, which as of 2023 covers 0.6 hectares and supplies produce to dining halls and stores on campus, local food banks, and CSA (Community Supported Agriculture) shareholders. Figure 1 shows the layout of the farm as of 2023, with the location of individual plots labeled. Table 1 reports the year during which each plot was added to the farm.

The UW Farm site at CUH has been farmed through a combination of organic and regenerative methods since Seattle Tilth began cultivation in 2002. As commonly defined,
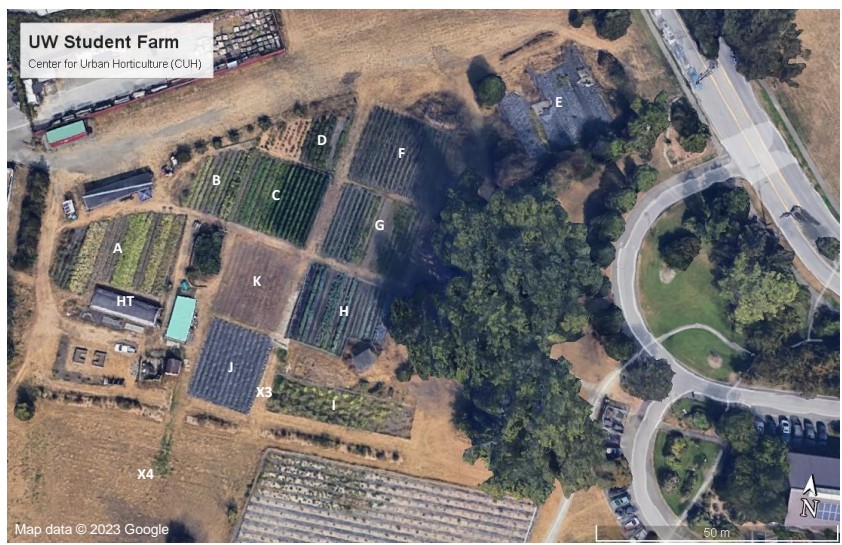

**Figure 1** **Map of UW Farm site in 2022, with plots labeled.** X3 and X4 mark locations of soil tests taken in 2022 for baseline, non-cultivated areas. Map data ©2023 Google.

**Table 1** **Data for soil plots.** UW Farm plots with the year the plot was incorporated to the farm and number of years under regenerative management practices, measured values for soil organic matter when available from 2018, 2019, 2021, and 2022, and topsoil depth measured in January or May 2023.

| Plot | Year added to farm | Years under regenerative management as of 2022 | SOM (%) May 2018 | SOM (%) May 2019 | SOM (%) April 2021 | SOM (%) Nov 2022 | Topsoil thickness (cm) 2023 |
|---|---|---|---|---|---|---|---|
| A | 2012 | 10 | 8.1 | – | 7.3 | 6 | 15.3 |
| B | 2011 | 11 | 5.5 | 5.9 | 7.4 | 4.9 | 15.8 |
| C | 2011 | 11 | – | – | 6.8 | 8.1 | 14.5 |
| D | 2011 | 11 | 7.1 | 7.7 | 7.8 | – | 16.0[*] |
| E | 1995 | 0 | 3.8 | 5.5 | 6.1 | 5.4 | 4.9 |
| F | 2011 | 11 | 8.4 | 14.9 | 8.1 | 8.2 | 14.5 |
| G | 2011 | 11 | – | 14.7 | 7.1 | 6 | 15.0 |
| H | 2002 | 20 | – | 13.1 | 18.3 | 25.1 | 20.4 |
| I | 2016 | 6 | – | 8.2 | 7.6 | 12.3 | 11.7[*] |
| J | 2017 | 5 | – | 8.3 | 6.8 | 4.0 | 8.6 |
| K | 2015 | 7 | – | 17.6 | 17.3 | 7.4 | 13.4 |
| HT | 2015 | 7 | 7.0 | 13.1 | 14.1 | 15.7 | 13.3[*] |
| X3 | – | 0 | – | – | – | 3.4 | 2.5[*] |
| X4 | – | 0 | – | – | – | 2.0 | 4.8[*] |

**Notes.**
*Topsoil depth measurements taken in May 2023.

**Table 2 Crop rotation order for a plot on the UW Farm.** Plant families are rotated in a set order annually throughout the plots both as a form of pest control and to reduce soil nutrient depletion. For example, a plot planted with crops from the Solanaceae family which consume large quantities of nitrogen will be planted the following year with crops from the Fabaceae family which host nitrogen-fixing bacteria in their roots and will restore soil nitrogen levels.

| Year | Plant family | Crops grown |
|------|--------------|-------------|
| 1 | Brassicaceae | Broccoli, cabbage, kale, collard greens, cabbage, kohlrabi |
| 2 | Solanaceae | Tomatoes, peppers, eggplants |
| 3 | Fabaceae | Peas, beans |
| 4 | Cucurbitaceae | Summer and winter squash, melons |
| 5 | Poaceae | Corn, wheat |
| 6 | Fallow | Planted with a cover crop (a mix of rye, barley, vetch, clover) and not used for growing produce |

both regenerative agriculture and organic agriculture aim to minimize disturbances, with the former focusing on minimizing physical disturbances and the latter on eliminating chemical disturbances. Although the UW Farm focuses on meeting requisites for USDA Organic and Good Agricultural Practices (GAP) certifications, and has been certified as USDA Organic since 2019, the methods used on the farm also focus on building soil health. The farm is minimal-till, occasionally utilizing a handheld tilther to mix compost into the top inch of soil when preparing beds at the beginning of the season, and annually utilizing a shallow BCS walk-behind tractor to seed cover crops beyond the reach of birds and other animals, but generally leaving the rest of the soil undisturbed. Compost is added to beds at a rate of approximately 0.7 kg m$^{-2}$ (7 t/ha) annually, including both commercially produced compost (Cedar Grove Compost) as well as vermicompost produced on site. Cover crops are used during fallow periods and winters. Plant families are rotated annually through the beds to disrupt pests and prevent soil nutrient depletion. Table 2 lists the typical crop rotation order and common crops grown on the farm. No synthetic chemical fertilizers or pesticides are used, and if needed soils are amended with natural products like blood meal, lime, kelp, and feather meal. Crops are often grown in a polyculture, with mutually beneficial plant species grown together in a row. Finally, cover crops and weeds are terminated using a tarping and solarization method and allowed to decay back into the soil.

When broadly defined as a collection of agricultural methods used to enhance soil and ecosystem health, the methods utilized on the UW Farm are an example of regenerative farming using Conservation Agriculture principles (*Kassam, Friedrich & Derpsch, 2019*) that has been implemented on site in a consistent manner since 2002. Plot E was cultivated in 1995 using more conventional methods including tillage, and while it has been occasionally weeded, tarped, and kept clear of brush, the area has not been regeneratively cultivated and therefore serves as a reference site. In addition, samples X3 and X4 were taken from a bare dirt pathway and uncultivated field respectively and also represent an initial baseline for the evolved landfill cap without agricultural management. A space-for-time analysis of the various UW Farm plots can be used to assess changes in soil development, with plots

E, X3, and X4 serving as baseline plots and plots A-D and F-K representative of soil under 5 to 20 years of regenerative cropping.

## METHODS

Topsoil thickness measurements were taken at three random locations across each plot. Soil pits were dug to a depth of approximately 60 cm and topsoil thickness measured as the distance from the ground surface to the base of the visually identified soil A-horizon. SOM values were retrieved from soil tests performed in May 2018, May 2019, April 2021, and November 2022. For each tested plot 10 soil samples were taken to a depth of approximately 25 cm and aggregated into a single sample sent to A&L Western Agricultural Laboratories for analysis. The lab used a standard loss-on-ignition test to determine % SOM. A conversion factor of 0.58 was used to convert % SOM to % SOC.

## RESULTS

The data show soil organic matter (Fig. 2; Table 1) and topsoil thickness (Fig. 3; Table 1) increased with time under regenerative management. Notably, there was a more than 2-fold increase in SOM between the baseline areas and the 20-year cultivated area, increasing from approximately 2 to 5% initially to >13%. Figure 2 shows the weak overall trend in SOM, increasing by approximately 0.5% per year, though with substantial variance and at the margin of statistical significance ($R^2 = 0.25$; $p < 0.07$). Additionally, there was roughly a 4-fold increase in average A-horizon thickness over the same period (Fig. 3), increasing from approximately 5 to 20 cm with a growth rate of about 0.86 cm per year ($R^2 = 0.93$; $p < 0.01$). Table 1 includes values for SOM when available from 2018, 2019, 2021 and 2022, and average topsoil thickness in 2023 for each plot, as well as the amount of time the plots have been managed regeneratively as of 2022 (when the most recent set of SOM measurements were taken).

Assuming an average top soil bulk density ($\rho$) of 1.0 g/cm$^3$ and a conversion factor of 0.58 between soil organic matter and soil organic carbon, an estimate of the total carbon storage in the topsoil of each plot can be determined from:

Equation 1: $OCS = 0.58 * SOM * \rho * d$

where OCS is organic carbon storage (t/ha), SOM is soil organic matter (%), and $d =$ topsoil (A horizon) thickness (cm). Figure 4 presents the trend in estimated organic carbon storage of the plots *versus* their time under regenerative management, showing OCS increasing from an initial average of about 15 tC ha$^{-1}$ to 300 tC ha$^{-1}$ on the oldest plot. While a linear regression ($R^2 = 0.79$) suggests an average annual increase of almost 10 tC ha$^{-1}$ yr$^{-1}$ , the minimum and maximum bounds on the data correspond to annual increases of 4 tC ha$^{-1}$ yr$^{-1}$ and 14 tC ha$^{-1}$ yr$^{-1}$ respectively. Only SOM data from 2022 were used to develop Fig. 4, as no prior historical measurements of topsoil thickness are available.
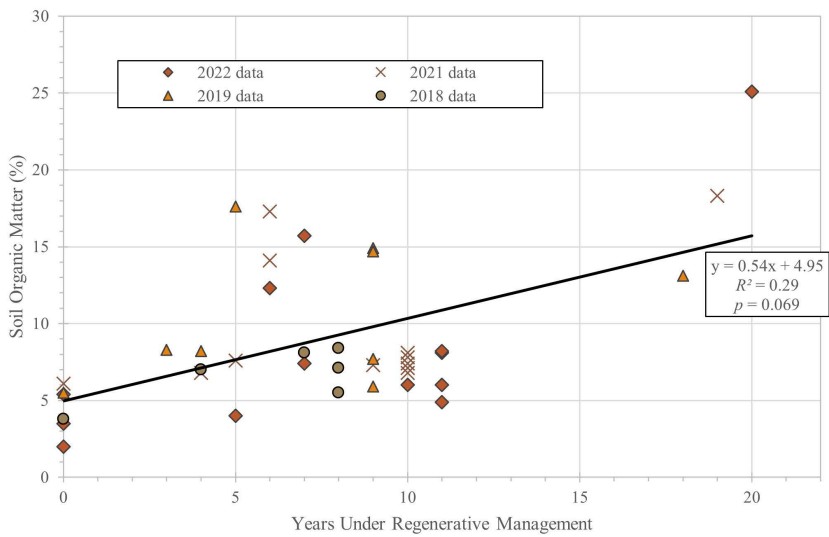

**Figure 2 Soil organic matter (%) *versus* years under regenerative management.**

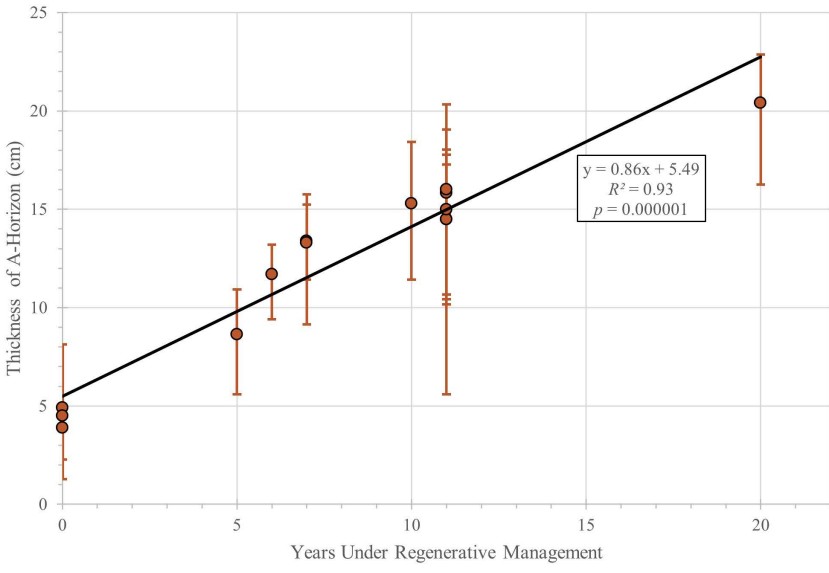

**Figure 3 Average topsoil depth (cm) *versus* years of regenerative management.** Error bars extend to maximum and minimum measured depths of A horizon for each plot.

## DISCUSSION

Our data show that both soil organic matter and topsoil thickness increased with time under regenerative management. In particular, the A-horizon on the UW Farm thickened at a rate of about 0.86 cm per year. The resulting four-fold increase in topsoil thickness on the UW Farm over the course of 20 years of regenerative management highlights the potential of regenerative agricultural methods to rapidly rebuild the fertility of degraded
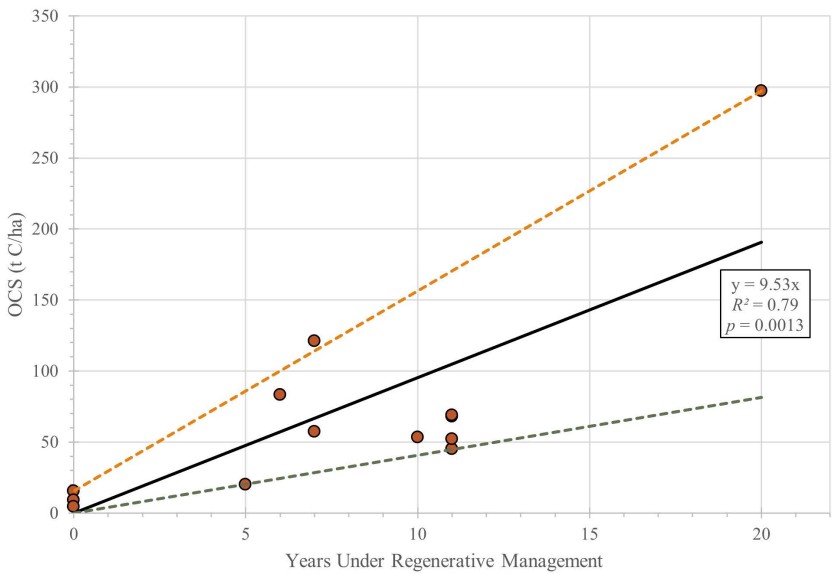

**Figure 4** **Estimated organic carbon storage (OCS) in topsoil (t C ha$^{-1}$) *versus* years managed regeneratively.** A linear regression (solid line; $R^2 = 0.79$, $p = 0.0013$) is shown, as well as maximum (orange dashed line; $y = 14.1x + 15.4$) and minimum (green dashed line; $y = 4.1x$) bounds to the data envelope.

soils. Natural topsoils, unaided by anthropogenic activity, are commonly estimated to increase in thickness by approximately a centimeter or two every 500 years, although rates vary depending on climatic and environmental parameters including temperature, moisture levels, and organic matter availability. The highest natural rate of soil formation reported to date is roughly 1 mm yr$^{-1}$ (*Larsen et al., 2014*). Hence, the observed rates of topsoil growth on the UW Farm outpace estimates of natural rates of topsoil formation by more than an order of magnitude, showing that restoration of degraded soil can build topsoil at rates that exceed geologic rates of topsoil production through rock weathering. In addition, the pace of soil building at the UW Farm exceeds values of 0.07 to 0.13 cm yr$^{-1}$ reported from trials of a routinely tilled farm with high biomass and additions of organic matter (*Bennett, 1939*; *Brady, 1984*). Topsoil growth rates on the farm also stand in marked contrast to trends of topsoil erosion on routinely tilled conventional farms (*Montgomery, 2007*; *Thaler, Larsen & Yu, 2021*), and are larger than rates reported by *Bennett (1939)* and *Brady (1984)*.

Overall, measured values for SOM increased over time, although the trend is more variable than the observed increase in topsoil thickness. While plots managed regeneratively all increased in SOM from the baseline samples, there was substantial year-to-year variability and individual plots did not always increase in SOM each year. Part of the variability is due to large additions of compost that temporarily increase SOM for a year or two. For example, in 2019 a large amount of composted coffee and leaves was added to plots F and K. Table 1 shows a high SOM value for the soils that year, and a decrease in the following years. Nonetheless, the overall trend of increasing SOM over time has significant implications for soil health and carbon sequestration. While much of the SOM regression is set by the

high levels of organic matter in plot H, even a growth rate of half that shown by the trend in Fig. 2 would result in substantial changes in SOM over decadal time scales.

Soil organic matter and topsoil thickness cannot increase indefinitely, however, and we would expect to see the growth rate for both soil organic matter and topsoil thickness slow and plateau over longer time scales (*Sauer et al., 2015*) as the soil system comes into equilibrium with new farming practices. Indeed, the structure of the regression residuals on Figs. 2 and 3 hints at an approach to such limits between 5–15% SOM and around 20 cm topsoil thickness. Hence, our data suggest that while regenerative practices like those employed on the UW Farm could serve as a rapid means of carbon sequestration over decadal time scales such rates should not be extrapolated over longer time scales.

While the Puget Sound region is not a moisture-limited environment, annually receiving approximately 1 m of precipitation and thus likely to have relatively rapid rates of soil development, the rate of organic matter increase shown in Fig. 2 is not without precedent. Two regenerative no-till vegetable farms studied by *Montgomery et al. (2022)* displayed an increase in SOM of between 7–10% over a decade or less of regenerative management, rates that exceed that of the general trend at UW Farm. Although SOM is not an ideal metric for soil health (which also depends on specific nutrient distributions as well as microbial and fungal diversity and abundance), it can act as a reasonable proxy in absence of other data. While increased SOM directly correlates to increased carbon content in soils, there is controversy over whether increasing SOM is an effective method of long-term carbon sequestration due to differences in the quality and stability of labile *versus* recalcitrant organic matter (*Lützow et al., 2007*).

The UW Farm SOM measurements provide minimum estimates for the topsoil (A-horizon) as they were obtained from an aggregate of 10 samples taken from across each plot, and extended to a depth of 25 cm and thus into the B horizon. Additionally, samples were taken by different student groups, and although they were all instructed and monitored by the same farm manager and tested by A& L Western Labs, there may have been minor variations in sampling process between groups and years. The observed changes in topsoil thickness integrate the organic-matter building effects of compost and mulch additions, root exudates, fungal action, and bioturbation by roots and earthworms.

The limited data available from the UW Farm show an increase in both topsoil thickness and soil organic matter over time. The compounded effect of these two changes can be used to estimate soil carbon storage; when combined with A-horizon thickening, the measured changes in SOM lead to estimated increases in overall carbon storage of between 4 and 14 t ha$^{-1}$ annually (Fig. 4). Considering that conventional agricultural practices often result in a net decrease in soil carbon, the demonstrated increase in soil carbon is significant. The range of estimated growth trends shown in Fig. 4 reflects the limited quantity of data available, as well as the relatively short amount of time the farm has been in operation. Note that extrapolating the trends in SOM and topsoil thickness to other farming systems or beyond the 20-year period of observations is questionable, as such growth is unlikely to be sustained, with values expected to plateau at new equilibrium levels. Additionally, we would expect these SOM values and rates of increase to be higher than for large-scale regenerative row cropping where importing compost is not feasible and cover crops and

crop residues provide the only source of additional organic matter inputs. Nonetheless, the changes observed on multi-decadal timescales on the UW Farm highlight the potential for small-scale urban farms to rapidly build soil carbon content while maintaining a productive agricultural environment.

## CONCLUSION

The UW Farm provides a novel space-for-time analysis of changes in soil under regenerative agricultural practices, displaying a twenty-year evolution of topsoil thickness and soil organic matter content. Data collected from 14 plots across the farm show an increase in both topsoil thickness and soil organic matter over time that all together illustrate an overall increase in soil carbon across farm plots under regenerative agriculture. In short, the UW Farm rapidly built topsoil and increased soil carbon while maintaining a productive growing space. The potential of regenerative farming methods to reduce soil erosion and increase soil organic matter and carbon spotlights their importance in future agricultural and climate-related policy. Further long-term analysis of the effects of regenerative agriculture on topsoil thickness and carbon content, as well as field trials that isolate individual methods and incorporation of more detailed soil analyses are needed to evaluate potential effects of wide-scale implementation of regenerative methods in different climatic settings and under various fertilization regimes. Finally, the UW Farm is largely operated by students and volunteers with little to no agricultural experience, and yet efficiently produces a large quantity of fresh produce, while simultaneously increasing soil volume and health. It is an example of a small, local, regenerative farm that showcases the potential for sustainable food production in urban environments.

## ACKNOWLEDGEMENTS

UW Farm Manager Perry Acworth and Eli Wheat graciously shared soil test data and their knowledge of management practices on the farm.

### Funding
This work was supported by the Department of Earth & Space Sciences at the University of Washington. The funders had no role in study design, data collection and analysis, decision to publish, or preparation of the manuscript.

### Grant Disclosures
The following grant information was disclosed by the authors:
The Department of Earth & Space Sciences at the University of Washington.

### Competing Interests
The authors declare they have no competing interests.

## Author Contributions

- Julia E. Macray conceived and designed the experiments, performed the experiments, analyzed the data, prepared figures and/or tables, authored or reviewed drafts of the article, and approved the final draft.
- David R. Montgomery conceived and designed the experiments, analyzed the data, authored or reviewed drafts of the article, and approved the final draft.

## Data Availability

The raw data are available in Table 1.

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
