# Peer review of "Trends in soil organic matter and topsoil thickness under regenerative practices at the University of Washington student farm"

_PeerJ, doi:10.7717/peerj.16336_

## Round 0.1 · original submission · Minor Revisions

In accordance with the reviewers’ opinions, I consider the manuscript scientifically sound and well written, and recommend minor revisions. Please take into account the recommendations of the reviewers and my own, included in the annotated manuscript.

·

Basic reporting

The paper is well written in a professional style, structured as scientific research paper and contains all necessary information as well as the most relevant literature.
It presents a clear objective and comes to a result based on own research.

Experimental design

The experimental design is adequate and well described. The research question is clear and the methods and statistical treatment explained.

Validity of the findings

The findings are well explained, validated against literature and clear. Corresponding data are presented. The conclusions are clear and derived from the data and supporting literature.

Additional comments

The paper addresses an important issue which is gaining international attention. It can also contribute to clarify actual controversies around carbon cycles and the role of regenerative agriculture.
Following some specific comments for the authors to consider:
Comments by line:
32-62: In the introduction it would be beneficial to mention, that the three “practices” mentioned as the core of regenerative agriculture reflect the three principles of “Conservation Agriculture”, which is a well defined and researched concept proving that only in combination these three principles produce regenerative results. This would provide a sounder background to the statements made in the introduction and also help better defining regenerative agriculture. In addition it would help promoting the term Conservation Agriculture, which is really the functional core of regenerative agriculture and with this help to get regenerative agriculture out of the area of being a fashionable buzz-word into a promising concept for sustainable land management. There is also a definition for regenerative agriculture available from the 1980s coming from Rodale Institute and more or less reflecting on the above. It might be beneficial to add this definition to the introduction.
107: cultivation techniques as a term always has the connotation of tillage. Better use a different, more neutral term, like cropping.
168-170: in reality the trend of declining growth rates and saturation can already be seen in the graph in figure 3, if the dots were not underlaid with a linear regression curve. Therefore, check whether the regression is really linear.
223: the term “regenerative practices” is not correct: the practices in isolation are not necessarily regenerative, but only in combination they result in regenerative results. Therefore, it is suggested to refer to Conservation Agriculture in the introduction and to use regenerative agriculture instead of regenerative practices if the entire package is referred to.

·

Basic reporting

Review Comments on PeerJ submission #90075

The paper provides a unique opportunity to identify changes on soil properties over a defined time and across different soil management practices. The experimental design, and discussions on methods is about as good as could be expected given that this is not a controlled experiment. Given this, the paper is suitable for publication, but with the following caveats:
1. The paper reports measurements in both metric and English units, e.g. lines 111-114, lines 116, etc. All measurements should be reported in metric.
2. The manuscript reports treatments for Regenerative Agriculture, but with no mention on how Regenerative Agriculture relates to the principles of Conservation Agriculture. Some discussion on this should be made to illustrate that the authors views are holistic, not myopic.
3. The authors use the terms soil depth and soil thickness interchangeably, and this causes confusion. THIS IS A MAJOUR ISSUE THAT NEEDS CLARIFICACTION. Also Table 2 does not state the units for depth.
4. The discussion on soil depth/thickness, or whatever, makes no mention if this is due strictly to depth of rooting, earthworms or other soil biota, or can other factors come into play. For example, the thickness of the A horizon can be caused by better rooting and/or surficial deposition of small quantities of loess (where rates of deposition are similar to rates of soil formation) over time, translocating soils from one place or another, etc. The authors assume the soil surface is definitive over time, but this is rarely the case. The authors should discuss other potential causes of soil depth/thickness to ensure that the entire picture has been considered.

Experimental design

Stated above.

Validity of the findings

Stated above

Additional comments

Stated above

---

## Round 0.2 · accepted · Accept

The authors have satisfactorily addressed the reviewers' comments and mine own. The article constitutes, in my opinion, a significant contribution to agricultural knowledge.

·

Basic reporting

The manuscript has been changed according to the review. No further comments.

Experimental design

no further comments

Validity of the findings

no further comments

Additional comments

The authors have amended the manuscript according to the reviewer comments. The manuscript is good to go.